# The Correlation between Chronic Endometritis and Tubal-Factor Infertility

**DOI:** 10.3390/jcm12010285

**Published:** 2022-12-29

**Authors:** Yujie Zou, Saijiao Li, Lei Ming, Yan Yang, Peng Ye, Jinjing Zou

**Affiliations:** 1Reproductive Medicine Center, Renmin Hospital of Wuhan University, Wuhan 430060, China; 2Gynecology Department, Third Hospital of Wuhan City, Wuhan 430060, China; 3Department of Pharmacy, Renmin Hospital of Wuhan University, Wuhan 430060, China; 4Department of Pulmonary and Critical Care Medicine, Renmin Hospital of Wuhan University, Wuhan 430060, China

**Keywords:** chronic endometritis (CE), tubal factor infertility, hydrosalpinx, inflammatory factors, plasma cell, laparoscopy

## Abstract

Objective: To identify the prevalence and risk factors for chronic endometritis (CE) with tubal factors and the correlation between chronic endometritis and tubal factors among infertile populations. Method: A total of 52 patients with chronic endometritis (CE group) who underwent laparoscopy and hysteroscopic surgery were recruited between July 2020 and December 2021. A total of 38 patients without chronic endometritis (non-CE group) were included as a control. Patients with endometriosis and intra-uterine abnormalities were excluded. Endometrial samples were collected during surgery for CD138 immunohistochemistry staining for the diagnosis of CE. Preoperative information (including age, reproductive health characteristics, previous medical and surgical history), intra-operative information (including the patency of the fallopian tube, the presence of hydrosalpinx, score and the grade of tubal lesion condition) and post-operative information (counts of CD138-positive HPF in the endometrial specimen) were collected. Result: A multivariate analysis revealed that tubal factors with unilateral or bilateral occlusion were significantly higher in the CE group (OR 3.066, 95% CI 1.020–9.213, *p* = 0.046). The bilateral occlusion of fallopian tubes (OR 8.785, 95% CI 1.408–54.818, *p* = 0.020) rather than unilateral occlusion (OR 2.860, 95% CI 0.893–9.162, *p* = 0.077) was significantly associated with chronic endometritis. The presence of a hydrosalpinx on one side (OR 7.842, 95% CI 1.279–48.086, *p* = 0.026) or both sides (OR 9.450, 95% CI 1.037–86.148, *p* = 0.046) was significantly associated with chronic endometritis. The comparison of CD138-positive HPF counts among the tubal occlusion patients without hydrosalpinx, patients with unilateral hydrosalpinx and patients with bilateral hydrosalpinx were as follows: 1 HPF (50.00% vs. 12.50% vs. 11.11%, *p* = 0.051), 2 HPF (38.89% vs. 25.00% vs. 22.22%, *p* = 0.615), ≥3 HPF (11.11% vs. 62.50% vs. 66.67%, *p* = 0.005). The stage of tubal condition was positively correlated with CD138-positive HPF counts in women with chronic endometritis (r = 0.460, *p* = 0.001). Conclusion: CE was closely related to the blockage of fallopian tubes and hydrosalpinx. The severity degree of the fallopian lesion condition was associated with inflammation of the endometrium.

## 1. Introduction

Chronic endometritis (CE) is considered a persistent local inflammation with an over-exudation of plasma cells infiltrated in the endometrial interstitial [1,2]. It is well-recognized that chronic endometritis is detrimental to female reproductive health, which could impact the process of embryo implantation and decrease pregnancy rates [3]. The prevalence of CE among the female population of reproductive age varied from 2.8% to 56.8% in infertile couples [4,5]. This was possibly due to the different diagnostic standards chosen. With advances in molecular biology, the identification of plasma cells with endometrial biopsies and immunohistochemistry staining (IHC) was gradually considered to be the gold standard for diagnosing CE. CD138, a trans-membrane heparan sulfate proteoglycan syndecan-1, was well-recognized to be the most specific molecule to identify the IHC of plasma cells for CD138, displaying an advantage in qualifying and quantifying CE [5]. Other methods, including hysteroscopy and uterine microbial detection, were also used for the evaluation of CE, while the IHC of plasma cells by CD138 was shown to have better objectivity and reliability. Different from acute endometritis, patients with chronic endometritis could show no evident clinical manifestation or present with just subtle discomfort, including pelvic pain, dysmenorrhea and abnormal leucorrhoea.

The pathological reason for chronic endometritis remained unclear, and microbial infection and disrupted immune micro-environment inside the uterus cavity were generally recognized to be responsible for this. A series of common bacteria were reported in the endometrium with chronic endometrium, including Mycoplasma genitalium, Escherichia Coli, Mycoplasma hominis, Enterococcus faecalis, fungi, etc. [6]. The related risk factors of chronic endometritis were complex. Egbe et al. [7] reported that a history of Chlamydia trachomatis and Mycoplasma infection was found to be an independent risk factor for tubal infertility. Multiple risk factors by Chen et al. [5] were analyzed, including delivery history, previous abortion history, history of spontaneous abortion, history of vaginitis, history of pelvic inflammation, history of cervical chlamydial infection and tubal blockage.

Tubal factor infertility was one of the most frequent reasons for female infertility, which accounted for 25–40% of female infertility. Tubal damage possibly involved the proximal or distal part or the entire fallopian tube. It was reported that about 80% of tubal damage occurred in the distal part frequently, and hydrosalpinx was the most severe manifestation, while in 10–25% of cases, the proximal part of the fallopian tube was affected [8]. Limited studies reported the correlation between tubal disease and the occurrence of chronic endometritis. Peng et al. [9] found that the presence of hydrosalpinx increased the incidence of CE, while bilateral hydrosalpinx did not significantly increase the incidence of CE compared with unilateral hydrosalpinx. Holzer et al. [10] reported that unilateral/bilateral tubal blockage (*p* = 0.013) and endometriosis (*p* = 0.034) were associated with chronic endometritis in a prospective study that included women with tubal factors and/or endometriosis. In addition to tubal disease, these two studies recruited patients confounded with other diseases, e.g., endometriosis and intra-uterine abnormalities, which may make the corresponding analysis complicated and introduce bias to the final conclusion. Moreover, the quantified analysis between CE and tubal factor disease was missing in these two studies. Given the advances in the diagnostic standards of CE and the evaluation system of tubal conditions, it would be interesting to know about CE’s manifestation and severity degree when these advances are applied to the tubal factor population nowadays. Thus, to be precise and convincible, it is necessary to conduct the study with the confounding conditions excluded and to quantify the correlation analysis between the severity of the tubal lesions and CE in our study.

In our center, we adopted the criterion raised by Chen et al. [5], namely, an immunohistochemical staining finding of ≥5 plasma cells per high-power field was considered CE positive. In order to calculate the minimum number for the immunohistochemical analysis of CD138+ plasma cells and identify a clinically relevant CE, 716 patients were retrospectively analyzed and divided into two groups: the CD138^low^ group (<5 CD138+ cells in all HPFs) and the CD138^high^ group (≥5 CD138+ cells in at least one HPF). It was revealed that the β-hCG positive rate (*p* = 0.05), clinical pregnancy rate (*p* = 0.01), and live birth rate (*p* = 0.02) were significantly lower in the CD138^high^ group than in the CD138^low^ group. Li et al. [11] drew the conclusion that the immunohistochemical analysis of CD138+ cells was a reliable method to detect CE, which can be identified by the presence of ≥5 plasma cells in at least one out of thirty HPFs, which provided the theoretical basis for the diagnostic standard adopted in our study.

Based on the communicable characteristics between the uterus cavity and fallopian tubes, we hypothesized that tubal factors, including hydrosalpinx, were correlated with the occurrence of chronic endometritis and inflammation inside the uterus. With endometriosis and uterine abnormality disease excluded, we conducted a retrospective cohort study among infertile patients with CE and without CE, aimed at exploring whether tubal factors alone were a risk factor for chronic endometritis by multivariate logistic analysis. To further investigate the relationship between chronic endometritis and the severity of tubal factors, we evaluated the lesion condition of the fallopian tube by the scoring system and analyzed the correlation between the lesion stage of tubal conditions and counts of CD138-positive HPF.

## 2. Materials and Methods

### 2.1. Patients

This retrospective study was conducted in the Reproductive Medicine Center, Renmin Hospital of Wuhan University, between July 2020 and December 2021.

The inclusion criteria were as follows: (1) patients met the diagnostic standard for infertility (couples lived together with a normal sex life for one year and were unable to sustain a pregnancy without contraception); (2) aged 20 to 38 years old; (3) underwent laparoscopy with chromopertubation as part of the infertility evaluation combined hysteroscopy with endometrial biopsy.

The exclusion criteria were as follows: (1) history of removal of one or two fallopian tubes; (2) the presence of endometriosis and adenomyosis; (3) the presence of uterine abnormalities, including endometrial polyps, intra-uterine adhesions, septate uterus, myomas and cesarean scar syndrome; (4) the presence of tuberculosis; (5) historical use of oral contraceptive pills, estrogen or progestin; (6) a medical history of endocrine disorders, including polycystic ovary syndrome (PCOS), diabetes mellitus and thyroid diseases; (7) use of antibiotics agents within three months. The separate diagnosis of endometriosis, adenomyosis and uterine abnormalities was confirmed based on the symptoms, physical examinations, laboratory testing and intra-operative findings, which met the corresponding diagnostic standard [12,13,14,15,16].

A total of 90 patients were finally included and divided into two groups according to the results of the immunohistochemical staining: the CE group (*n* = 52) and the non-CE group (*n* = 38) (Figure 1).

All the approvals for the study were obtained from the Ethical Committee of Renmin Hospital of Wuhan University (Approval notice number: WDRY2020-K060), and written consent was obtained from all participants.

### 2.2. Surgical Procedure: Laparoscopy, Hysteroscopy and Endometrial Sampling

All subjects underwent hysteroscopy and laparoscopic exploration in the proliferative phase under general anesthesia. The hysteroscopy was performed with a 2.9-mm 30 rigid hysteroscope (Karl Storz, Tuttlingen, Germany) with an outer diameter of 4.3 mm. Normal saline solution was used to distend the uterine cavity at 100 mm Hg pressure. When the hysteroscopy was finished, the endometrial sample was gently obtained with a Pipelle biopsy catheter (Jiadingcheng, Guangdong, China) from the upper uterine cavity.

For the laparoscopy (Karl Storz, Germany), the internal reproductive organs and pelvic cavity were thoroughly inspected. In order to evaluate the patency of bilateral fallopian tubes, a laparoscopic chromopertubation was performed using a 50-mL syringe with a dilute solution of methylthioninium chloride (JUMPCAN, Taixing, China) through the uterine manipulator. The lesion condition of the fallopian tubes was observed and scored according to the recently developed tubal classification system by Zou et al. [17]. Four aspects were recorded: (1) patency (patent/passable/sub-passable/completely blocked); (2) morphology (soft/partially narrowed or twisted/stiffness or hydrosalpinx); (3) fimbrial structure (complete/partially destroyed/destroyed); (4) nature of adhesion (membranous/dense); (5) adhesion range (peritubal/partial ovarian/completely wrapped or extensive). The tubal lesion condition was accordingly graded as follows: grade 1, mild (0–7 points); grade 2, moderate (8–15 points); grade 3, severe (≥16 points).

For the following treatment, patients who were identified with pelvic adhesions during the laparoscopic exploration underwent pelvic adhesion lysis surgery. Patients who were identified with twisted or wrapped fallopian tubes underwent fimbrioplasty for the repair of the fallopian tube (*n* = 23 tubes), and those with fimbriae atresia or hydrosalpinx underwent neosalpingostomy (*n* = 27 tubes). Patients with proximal occlusion of the fallopian tubes underwent hysteroscopic tubal intubation combined with guide-wire intervention (*n* = 13 tubes). Patients with severe hydrosalpinx, which was identified with a total absence of physiological tubal function, underwent a salpingectomy (*n* = 2 tubes) or sterilization (*n* = 1 tube) after agreement.

### 2.3. Histological Staining

The endometrial sample was fixed in 10% formaldehyde, embedded in paraffin and later sliced into five-micrometer sections. HE staining and routine immunohistochemical staining were carried out as follows [18]. The immunohistochemical staining was performed by incubation with a 1:100 dilution of mouse anti-human monoclonal CD138 antibody (MI 15; Dako, Glostrup, Denmark) for 1 h at room temperature. The CD138-positive plasma cells were observed in the endometrial stroma and quantified under a light microscope (Olympus, Tokyo, Japan) at a magnification of 400×.

### 2.4. Diagnosis of Chronic Endometritis

At least 30 high-power fields (HPF) were examined per specimen. The immunohistochemical staining finding of ≥5 plasma cells per high-power field was considered CE positive according to the published criteria [5] and considered to be one positive HPF. The presence of 1–4 CD138-positive plasma cells or the absence of plasma cells was considered CE negative. All endometrial IHC specimens were evaluated by two histopathologists.

### 2.5. Measurement

The preoperative information was obtained from the Donghua outpatient management system in our center, including:(1)Age;(2)Reproductive health characteristics: anti-Müllerian hormone (AMH), body mass index (BMI), infertile duration, primary/secondary infertility;(3)Previous medical and surgical history: history of curettage, pelvic inflammatory disease, intra-uterine contraceptive device use, genital infection (Chlamydia trachomatis, Mycoplasma), abdominal surgery.

Pelvic inflammatory disease (PID) was defined as a history of self-reported episodes of inflammation in the pelvic cavity and female reproductive system and diagnosed by a professional clinician. In order to obtain more knowledge, women were further questioned whether the diagnosis was based on reported symptoms, physical examination or laboratory testing (blood or vaginal swab examination) or whether they were admitted to the hospital for the PID episode.

Genital infection (Chlamydia trachomatis, Mycoplasma) was defined as a positive outcome with a culture for Chlamydia trachomatis and Mycoplasma using the high vaginal swab or a positive PCR test outcome and/or a self-reported Chlamydia trachomatis/Mycoplasma infection.

The intra-operative information was collected, including the following: patency of the fallopian tube, hydrosalpinx presence and the score and grade of the tubal lesion condition for each subject following Section 2.2.

The post-operative information on the endometrial specimen was collected, including counts of CD138-positive HPF.

### 2.6. Statistical Analysis

Continuous variables were presented as mean values ± SD and analyzed by using independent *t*-tests among the CE and non-CE groups. The Chi-square test was used to compare the categoric variables. Logistic regression models were fitted after an adjustment for age, reproductive health characteristics (AMH, BMI, infertile duration, primary/secondary infertility), and medical and surgical factors (history of curettage, pelvic inflammatory disease, intra-uterine contraceptive device use, genital infection, abdominal surgery) associated with chronic endometritis. The crude odds ratios by unconditional logistic regression, adjusted odds ratios (aOR) by multivariate logistic regression models and corresponding 95% confidence interval (95% CI) were interpreted. Correlations between the stage of the tubal condition and corresponding CD138-positive HPF counts were calculated with the Spearman rank correlation coefficient. The statistical analysis was conducted using SPSS 17.0, and a *p* < 0.05 was considered statistically significant.

## 3. Results

### 3.1. Comparison of General Information and Risk Factors between CE and Non-CE Group

As shown in Table 1, the background clinical characteristics were compared between the CE group and the non-CE group. No significant difference was observed in age, AMH, BMI, infertile duration and the type of infertility between the two groups. As for the comparison of information with previous medical and surgical history, no significant difference was found in the history of curettage, pelvic inflammatory disease, intra-uterine contraceptive device use and abdominal surgery between the two groups, while the prevalence of genital infection (Chlamydia trachomatis, Mycoplasma) was found to be significantly higher in the CE group than in the non-CE group (26/52 = 50.00% vs. 10/38 = 26.32%, *p* = 0.023). As for the comparison of tubal factor infertility, the presence of a unilateral or bilateral occlusion was found to be significantly higher in the CE group than in the non-CE group (35/52 = 67.31% vs. 15/38 = 39.47%, *p* = 0.007).

### 3.2. Logistic Regression Analysis for CE Associated with Covariates

Logistic regression models were fitted after an adjustment for age, reproductive health characteristics, and the medical and surgical factors associated with chronic endometritis. As shown in Table 2, estimates of the risk of chronic endometritis for different variables are summarized. In the multivariate logistic analysis, a history of a genital infection (OR 3.356, 95% CI 1.219–9.237, *p* = 0.019) was strongly associated with chronic endometritis. No significant difference was found with respect to a history of curettage (OR 1.087, 95% CI 0.405–2.912, *p* = 0.869), pelvic inflammatory disease (OR 1.560, 95% CI 0.453–5.380, *p* = 0.481), intra-uterine contraceptive device use (OR 1.027, 95% CI 0.190–5.555, *p* = 0.975), and abdominal surgery (OR 1.036, 95% CI 0.200–2.706, *p* = 0.645) between the CE group and non-CE group. When adjusted for the effects of age, reproductive health characteristics (AMH, BMI, infertile duration, primary/secondary infertility), and medical and surgical factors (history of curettage, pelvic inflammatory disease, intra-uterine contraceptive device use, genital infection, abdominal surgery), tubal factors were significantly correlated with the occurrence of chronic endometritis (OR 3.066, 95% CI 1.020–9.213, *p* = 0.046).

As shown in Table 3, by a multivariate binary regression model, the presence of a hydrosalpinx on one side (OR 7.842, 95% CI 1.279–48.086, *p* = 0.026) or both sides (OR 9.450, 95% CI 1.037–86.148, *p* = 0.046) was significantly associated with chronic endometritis. Interestingly, the bilateral occlusion of the fallopian tubes (OR 8.785, 95% CI 1.408–54.818, *p* = 0.020) rather than unilateral occlusion (OR 2.860, 95% CI 0.893–9.162, *p* = 0.077) was significantly associated with chronic endometritis.

### 3.3. Correlation between CD138-Positive HPF Counts and Severity of Tubal Condition in Women with Chronic Endometritis

As the most severe manifestation of a tubal lesion, the correlation between hydrosalpinx and CD138-positive HPF counts is analyzed in Table 4. The comparison of the CD138-positive HPF counts among the tubal occlusion patients without hydrosalpinx, patients with unilateral hydrosalpinx and patients with bilateral hydrosalpinx are shown in Table 4: 1 HPF (50.00% vs. 12.50% vs. 11.11%, *p* = 0.051), 2 HPF (38.89% vs. 25.00% vs. 22.22%, *p* = 0.615), ≥3 HPF (11.11% vs. 62.50% vs. 66.67%, *p* = 0.005). Notably, no significant difference was found for the counts of CD138-positive HPF between patients with unilateral hydrosalpinx and patients with bilateral hydrosalpinx (*p* > 0.05). Interestingly, 88.89% of CE in the tubal occlusion patients without hydrosalpinx manifested as moderate CE (CD138-positive HPF counts: 1–2), while 62.5% of CE in patients with unilateral hydrosalpinx and 66.67% in patients with bilateral hydrosalpinx manifested as severe CE (CD138-positive HPF counts: ≥3). The comprehensive score of the tubal condition in each subject was recorded and classified: grade 1 (*n* = 19, mild); grade 2, (*n* = 14, moderate); grade 3, (*n* = 19, severe). The positive correlation between the stage of the tubal condition and corresponding CD138-positive HPF counts in women with chronic endometritis is shown in Figure 2 (r = 0.460, *p* = 0.001).

## 4. Discussion

There were three strengths in this study. First, to our best knowledge, using the newly developed scoring system for comprehensively evaluating tubal conditions, this study first demonstrated a positive correlation between the severity degree of fallopian lesions and inflammation of the endometrium in a quantified way. Second, different from previous research [5,9], patients with only tubal factors were recruited in this study; confounding diseases such as endometriosis and intra-uterine abnormalities were excluded to keep the analysis precise and convincible. Third, this study identified that the bilateral occlusion of fallopian tubes rather than unilateral occlusion was significantly associated with chronic endometritis. However, the presence of a hydrosalpinx on no matter one side or both sides was dependently associated with chronic endometritis. Interestingly, CE mainly manifested as a severe degree in patients with hydrosalpinx while manifesting as a moderate degree in tubal factor patients without hydrosalpinx, providing new insight into the correlation of CE and tubal factors.

Thus far, several evaluation methods, including hysteroscopy, immunohistochemical staining for plasma cells in the endometrial stroma and uterine microbial detection, have been used to identify chronic endometritis. In our center, we adopted the diagnostic criterion for CE raised by Chen et al. [5] and further validated by Li et al. [11], namely, the immunohistochemical staining finding of ≥5 plasma cells per high-power field in at least one out of thirty HPFs was considered CE positive. The immunohistochemical staining of the plasma cell surface antigen CD138 is well-recognized for both qualifying and quantifying CE, though there was an inconsistency about the exact number of plasma cells required in the endometrial stroma in different reports. To summarize and evaluate comprehensively among different diagnostic criteria, Huang et al. conducted a meta-analysis with twelve eligible studies involving 1879 patients [19]. Among the six different diagnostic criteria, from broad to strict, they were separate: ≥1 plasma cell per section; ≥1 plasma cell per 10 hpf; an endometrial stromal plasmacyte density index (ESPDI) ≥ 0.25; ≥1 plasma per hpf; 1–5 plasma cells per hpf or discrete clusters of <20 plasma cells; ≥5 plasma cells per high-power field (hpf). These two (≥5 plasma cells per hpf and ≥1 plasma cell per section) were the most commonly used.

The results of the logistic regression analysis by Chen et al. [5] confirmed that a fallopian tube obstruction (OR = 3.274, 95% CI 1.139–9.415, *p* = 0.028) was an independent risk factor for chronic endometritis. A total of 624 patients with hydrosalpinx and 789 patients without hydrosalpinx undergoing laparoscopy and hysteroscopy were retrospectively analyzed by Peng et al. [9], and they also found that hydrosalpinx displayed a significant impact on the prevalence of CE (plasma cell count: no plasma cells: OR 0.71, 95% CI 0.58–0.88, *p* = 0.002; ≥1/HPF: OR 1.40, 95% CI 1.14–1.74, *p* = 0.002; ≥3/HPF: OR 1.50, 95% CI 1.18–1.91, *p* = 0.001; ≥5/HPF: OR 1.62, 95% CI 1.27–2.21, *p* < 0.001), while no significant differences in the comparison of plasma cell counts were observed between the unilateral hydrosalpinx group and the bilateral hydrosalpinx group, which was in accordance with our findings. In our study, the comparison of the CD138-positive HPF counts among tubal occlusion patients without hydrosalpinx, patients with unilateral hydrosalpinx and patients with bilateral hydrosalpinx were as follows: 1 HPF (50.00% vs. 12.50% vs. 11.11%, *p* = 0.051), 2 HPF (38.89% vs. 25.00% vs. 22.22%, *p* = 0.615), ≥3 HPF (11.11% vs. 62.50% vs. 66.67%, *p* = 0.005), which varied from Peng’s study, possibly due to the different diagnostic criterion chosen and the limited sample size. As the most severe manifestation of a tubal lesion, hydrosalpinx was the distension or dilatation of the fallopian tube in the presence of a distal tubal occlusion and was accompanied by a generally recognized lower implantation and lower pregnancy rates. Interestingly, in our study, 88.89% of CE in tubal occlusion patients without hydrosalpinx manifested as moderate CE (CD138-positive HPF counts: 1–2), while 62.5% of CE in patients with unilateral hydrosalpinx and 66.67% in patients with bilateral hydrosalpinx manifested as severe CE (CD138-positive HPF counts: ≥3), indicating that hydrosalpinx, no matter on one side or both sides, was associated with a severe inflammation response in the endometrial environment compared with the tubal occlusion without hydrosalpinx.

As to the anatomical aspect of the female reproductive system, the uterine cavity was communicable with the pelvic cavity by the oviducts; thus, immune factors and pathogens could pass between the cavities. Some humoral factors or substances produced in the fallopian tube could return to the uterine cavity and may induce plasma cell infiltration in the endometrial stroma. Conversely, the over-exudative plasma cells could flow into the fallopian tube and pelvic cavity. In order to explore the impact of hydrosalpinx fluid on the endometrium, the expression of inflammatory cytokines TNF-a and IL-2 mRNA in endometrial tissue was detected. It was found that patients with hydrosalpinx had significantly higher expressions of IL-2 and TNF-α mRNA in their endometrial tissue, which played a crucial role in the inflammatory response and may therefore affect endometrial receptivity [20]. No matter whether the inflammation came from the lower reproductive tract or upper pelvic cavity, inflammation itself led to the impairment, or even absence of ciliated cells of the fallopian tubal endothelium, resulting in dysfunction of the fallopian tube and the detrimental transportation of gametes and embryos. Moreover, over-exudative inflammation factors may get involved in the development of intra-luminal, peri-tubal adhesion or occlusion of the fimbrial part. A proximal tubal obstruction could result from amorphous debris or mucus plugs, endometriosis, pelvic inflammatory disease and uterine polyps over the inner opening of the fallopian tube [10].

It was arguable that confounding diseases contributed to the development of chronic endometritis. According to the multivariate analysis by Kuroda et al. [12], the presence of endometrial polyps and intra-uterine adhesions significantly increased the risk of the development of CE, respectively, by 27.69 and 8.85 folds. The occurrence of endometrial polyps and intra-uterine adhesions was reported to be related to the chronic inflammation of the local endometrium, with an over-secretion of inflammation-related cytokines [13,14]. The histopathological detection of cesarean scar defect (CSD) lesions showed a significantly higher expression of the chronic inflammatory marker, CD138, while significantly lower expressions of acute inflammatory markers CD3, CD20 and CD68 in the cesarean scar syndrome (CSS) group were seen when compared with the non-CSS group [15]. It was revealed by Takebayashi et al. [16] that chronic endometritis was identified in 52.94% (18/34) of the endometriosis group and 27.02% (10/37) of the non-endometriosis group (*p* = 0.03). The logistic regression analysis revealed that endometriosis was strongly associated with chronic endometritis. It was reported by Holzer et al. [10] that the mean number of CD138-positive cells was positively correlated with the revised American Society for Reproductive Medicine score in women with endometriosis (r = 0.302, *p* = 0.028). Considering intra-uterine abnormality, including endometrial polyps, intra-uterine adhesion, cesarean scar syndrome and endometriosis as possibly constituted risk factors for chronic endometritis, these conditions were excluded when we recruited the subjects in this study in order to be precise and convincing.

Considerable data suggested that hydrosalpinx and chronic endometritis were harmful to embryo implantation and pregnancy outcome. There were many theories of the mechanism about whether direct embryotoxicity by mechanical and chemical factors—the inflammatory response inside the endometrial cavity—or suboptimal endometrial receptivity played a role [21]. The examination of tubal (106.0 ± 123.3 vs. 29.2 ± 22.7, *p* < 0.001) and endometrial slides (79.9 ± 133.4 vs. 34.9 ± 40.3, *p* = 0.03) with hydrosalpinx or salpingitis indicated a significant increase in the number of overall inflammatory cells when compared with the control [22]. A defined, identifiable, local response to hydrosalpingeal fluid has been demonstrated in the endometrium. This response comprised statistically significant elevations of leukocytes and IL-2 by high-intensity immunohistochemical staining (65% vs. 7.4%, *p* = 0.0001). An endometrial inflammatory response may be an independent contributor to the decreased reproductive outcome observed in patients with hydrosalpinges. Moreover, it was demonstrated that the HOXA10 expression was down-regulated in the endometrium of patients with hydrosalpinx, which adversely affected the attachment and implantation process [23].

Several tubal classification systems were used to assist in the reasonable evaluation of tubal conditions and following pregnancy outcomes, though no agreement was reached for the standard worldwide. The most popular ones included the revised American Society for Reproductive Medicine’s revised classification of endometriosis (AFS-r), the Hulka tubal classification system, the Hull and Rutherford classification system and falloposcopy while with separate limitations for their clinical application [24,25,26,27]. The AFS-r system was developed to be more suitable for evaluating pelvic adhesions and endometriosis. The Hull and Rutherford classification system evaluated the tubal condition to be good, general and poor, which was a descriptive way without using a scoring system. A falloposcopy was not often used and can only be applied in the diagnosis of the inner condition of fallopian tubes. Zou et al. [10], in 2014, developed a new system for evaluating the tubal condition and pregnancy prognosis based on the tubal adhesion range, tubal patency, tubal morphology and fimbrial structure. This was a combination of the AFS-r, Hulka system and the Hull and Rutherford classification. The retrospective data, including 1290 patients, showed significant association with intra-uterine pregnancy (mild: 43.6% vs. moderate: 34.0% vs. severe: 19.4%, *p* < 0.0001) and live births rates (mild: 35.9% vs. moderate: 31.5% vs. severe: 16.8%, *p* = 0.0002) using pre- and post-operative TFI classifications. The multivariate analysis showed that the preoperative disease course (*p* = 0.02), preoperative TFI score (*p* < 0.0001), and post-operative TFI score (*p* = 0.0007) were independently associated with the rate of intrauterine pregnancy rates. Thus, we adopted this new classification system in our study and considered it evidence-reliable.

This study has several limitations. First, it was a retrospective study, and the data was derived from a single-center experience; thus, relevant clinical information associated with CE could be missing. We used self-reported data on pelvic inflammatory disease and, in part, on genital infections (Chlamydia trachomatis, Mycoplasma), which may be subject to recall bias. Second, the sample size of our study was limited, which may constrain the power of the statistical tests to detect a significant difference; however, our study still provided some important clues for future direction. Third, the microbiological reason, type and level of inflammation factors were not detected; thus, the infectious and immunological mechanisms were not clarified in this study. Fourth, the strong correlation between chronic endometritis and tubal disease was convincing in this study; however, it was not demonstrated whether tubal disease would be responsible for CE or whether CE itself led to an inflammatory environment that caused a tubal blockage.

In summary, this study added to the evidence that CE was closely and independently related to the blockage of fallopian tubes and hydrosalpinx and also showed that the severity degree of fallopian lesion conditions was positively associated with inflammation of the endometrium. For infertile women with a bilateral occlusion of the fallopian tubes or with hydrosalpinx, clinicians should be aware of the potential risk of CE and may accordingly propose an endometrial biopsy plan for diagnosing CE to prevent implantation problems. The inner infectious or immunological mechanisms remained unclear, and further exploration is needed. In the future, prospective, randomized, controlled and multi-center clinical studies would be warranted to confirm the conclusion.

## Figures and Tables

**Figure 1 jcm-12-00285-f001:**
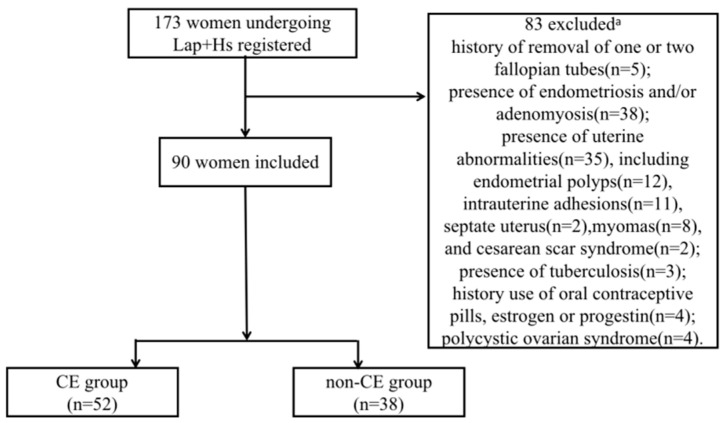
Data processing. ^a^ Some patients met more than 1 criterion for exclusion.

**Figure 2 jcm-12-00285-f002:**
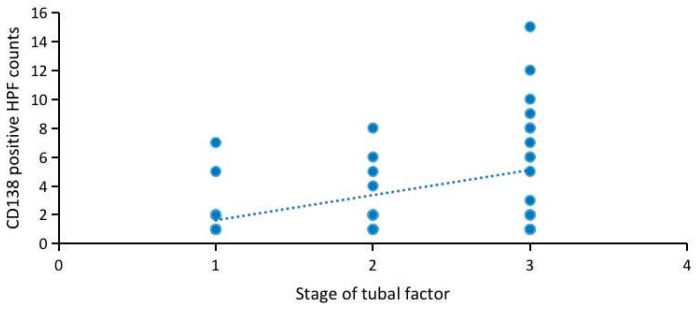
Correlation between CD138-positive HPF counts and stage of tubal factor in women with CE (*n* = 52).

**Table 1 jcm-12-00285-t001:** Comparison of general information and risk factors between CE and non-CE groups.

	CE Group (*n* = 52)	Non-CE Group (*n* = 38)	*p*-Value
Age (years)	30.54 ± 2.48	30.68 ± 2.79	0.795
AMH (ng/mL)	2.39 ± 0.63	2.52 ± 0.61	0.307
BMI (kg/m^2^)	20.75 ± 1.15	21.20 ± 1.04	0.061
Infertile duration (years)	1.88 ± 0.69	2.01 ± 0.61	0.131
Type of infertility			0.300
Primary infertility	22	12	
Secondary infertility	30	26	
Previous history of curettage			0.288
Yes	25	14	
No	27	24	
Previous history of pelvic inflammatory disease			0.210
Yes	14	6	
No	38	32	
Intrauterine contraceptive device			0.777
Ever used	5	3	
Never used	47	35	
Genital infection			0.023
Yes	26	10	
No	26	28	
Abdominal surgery			0.521
Yes	11	6	
No	41	32	
Tubal factors			0.007
Unilateral or bilateral occlusion	35	15	
Bilateral patency	17	23	

**Table 2 jcm-12-00285-t002:** Odds ratio and 95% confidential interval (CI) for CE associated with covariates.

Risk Factors	Crude Risk		Adjusted Risk		*p*-Value
	OR	95% CI	OR	95% CI	
Previous history of curettage					
Yes	1.587	0.675–3.731	1.087	0.405–2.912	0.869
No	1	/	1	/	/
Previous history of pelvic inflammatory disease					
Yes	1.965	0.677–5.704	1.560	0.453–5.380	0.481
No	1	/	1	/	/
Intrauterine contraceptive device					
Ever used	1.241	0.278–5.544	1.027	0.190–5.555	0.975
Never used	1	/	1	/	/
Genital infection					
Yes	2.800	1.134–6.913	3.356	1.219–9.237	0.019
No	1	/	1	/	/
Abdominal surgery					
Yes	1.431	0.478–4.285	1.036	0.200–2.706	0.645
No	1	/	1	/	/
Tubal factors					
Unilateral or bilateral occlusion	2.896	1.219–6.884	3.066	1.020–9.213	0.046
Bilateral patency	1	/	1	/	/

**Table 3 jcm-12-00285-t003:** Comparison of tubal factors between CE and non-CE groups.

	CE (*n* = 52)	Non-CE (*n* = 38)	aOR	95%CI	*p*-Value
Patency					
Bilateral patency	17	23	1	/	/
Unilateral occlusion	21	13	2.860	0.893–9.162	0.077
Bilateral occlusion	14	2	8.785	1.408–54.818	0.020
Hydrosalpinx					
No hydrosalpinx	35	35	1	/	/
Unilateral hydrosalpinx	8	2	7.842	1.279–48.086	0.026
Bilateral hydrosalpinx	9	1	9.450	1.037–86.148	0.046

**Table 4 jcm-12-00285-t004:** Correlation of hydrosalpinx and CD138-positive HPF counts.

	1 HPF	2 HPF	≥3 HPF
Tubal occlusion without hydrosalpinx	9/18 = 50.00%	7/18 = 38.89%	2/18 = 11.11%
Unilateral hydrosalpinx	1/8 = 12.50%	2/8 = 25.00%	5/8 = 62.50% ^a^
Bilateral hydrosalpinx	1/9 = 11.11%	2/9 = 22.22%	6/9 = 66.67% ^a^
*p*-value	0.051	0.615	0.005

^a^ Indicates a statistical difference when compared with tubal occlusion without hydrosalpinx (*p* < 0.05).

## Data Availability

The datasets generated during and/or analyzed during the current study are available from the corresponding author upon reasonable request.

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
