# Peer review of "The Correlation between Chronic Endometritis and Tubal-Factor Infertility"

_jcm, 2022, doi:10.3390/jcm12010285_

Round 1
Reviewer 1 Report
INTRODUCTION SECTION
In line 47 delete [et al] after fungi
please give a short note to the readers about CD138, its role in diagnosing CE, and its limitations.
define the gap in knowledge that earlier studies have missed and the rationale for conducting the current study.
lines 275-280 in the discussion are pre-elementary data for the study; we recommend moving them to the introduction.
in the METHOD SECTION AND RESULTS
exclusion criteria
were cases with PCOS syndrome, diabetic cases, thyroid diseases, or those on drugs like antibiotics excluded or not?
line 143 define abbreviation then use them
in line 193 and 195 how did you differentiate GENITAL INFECTION from CHRONIC PELVIC INFECTION?? according to which criteria and how did you confirm??
the figures for OR and the table represent the same info you should pick one rather than both.
DISCUSSION
YOU should highlight your study's main result
and mentions a reference to the first 10 lines of your discussion
line 274 -284 seems like the missing rationale of your study introduction??
line 307-317 are they your study data or others???
rephrase please in a more clear way.
line 347-353 what is relevant to your study aim??
what are the novel finding or the updates from your data ??
what is the home tacking massage? how are these data will impact the current knowledge and improver our management for tubal and CE treatment?
Reviewer 2 Report
I have evaluated your article about CE, tubal patency, hydrosalpenx. My observations are as follows:
First of all, the method part should be more descriptive and the way in which the exclusion criteria is determined should be specified. For example, how is the diagnosis of adenomyosis made?
Are hysteroscopy and laparoscopy performed on every infertile patient? The indication for the procedure may be associated with CE, especially RIF.
Figure 3 is also not clearly understood. Why was genital infection not considered a confounding factor in this regression analysis?
There are punctuation and spelling errors. I recommend developing the language of expression.
Round 2
Reviewer 1 Report
dear authors thank you for addressing most of the issues raised in our comments
we still have a few point that was not addressed yet
1. the flowchart seemed blurred and mixed with another picture. kindly check it and provide a clear flow chart
2. you promised to save either table or figure for the Odd ratio; now both are seen???
3. The home tacking massage and implication of the study need more tightening
Do you think that CE and tubal block may be linked to low endometrial receptivity?? see most of your patients are seeking pregnancy,having diagnosed the cause is truly half the way to cure; do you think that Endomateria scratching in these cases may of value?
